# Research on the Development of Religious Tourism and the Sustainable Development of Rural Environment and Health

**DOI:** 10.3390/ijerph18052731

**Published:** 2021-03-08

**Authors:** Hsiao-Hsien Lin, Ying Ling, Jao-Chuan Lin, Zhou-Fu Liang

**Affiliations:** 1Department of Leisure Industry Management, National Chin-Yi University of Technology, Taichung 41170, Taiwan; 2Department of Tourism Management, Athena lnstitute of Holidtic Wellness, Wuyi University, No 26, Wuyi Avenue, Wuyishan 354300, China; 3Institute of Physical Education and Health, Yulin Normal University, 1303 Jiaoyu East Rd., Yulin 537000, China; g169168@yahoo.com.tw; 4Department of Marine Leisure Management, National Kaohsiung University of Science and Technology, Kaohsiung 811213, Taiwan; jcl@nkust.edu.tw; 5School of Environmental and Life Sciences, Nanning Normal University, No. 175 Mingxiu East Road, Xixiangtang District, Nanning 530001, China

**Keywords:** god of wealth, cultural tourism, environmental awareness, environmental sensitivity, impact cognition

## Abstract

The purpose of the research is to explore how to reach a consensus on the development of cultural tourism and the sustainability of the entire rural environment from the perspective of different rights holders. Using Beigang Township in Taiwan as a case study, we first conducted a questionnaire survey and analyzed 600 respondents by statistical verifications method, then used an interview method to compile suggestions from experts and scholars, and finally conducted a field survey to collect actual information. After summarizing, organizing, and analyzing all the data, the study was examined in a multivariate manner. This study concludes that creating parking spaces, providing a comfortable resting place, facilitating the exchange of ideas, and improving the environmental literacy of the public will increase the public attention to issues such as village visibility, people interaction, ancient architecture, culture and totems, public health and transportation, and entrepreneurial development, as well as address the concerns of local residents and some men and people over 31–40 years old. By doing so, we can improve community building and security, enrich cultural resources, build and develop sufficient industries, stabilize prices, obtain a safe and hygienic village environment, increase the desire to revisit, become a recommendation for family travel, and achieve the goal of sustainable development of rural environment and health.

## 1. Introduction

Cultural tourism has gradually become an important source of income for the tourism industry. It is also a tourism asset that countries are investing in and developing one after another. Sites, architecture, art, festivals, religions, pilgrimages, etc., cultural relics or behaviors that can be remembered can be called cultural tourism resources [1]. However, general tourism resources will gradually be consumed due to the time and degree of use [2], and maintenance cost is required. However, religious beliefs and culture will not be exploited and consumed for development purposes, resulting in the exploitation or depletion of cultural resources that are increasingly impure. Instead, because of the uniqueness of local religious beliefs and culture, they are recognized by the public and attract more believers to worship them [3], which in turn adds to the mystery of local culture and makes local religious beliefs and culture more valued and preserved by the public [4]. It can be seen that religious cultural tourism resources are sustainable and have considerable potential for improving the current situation of rural development.

Religious culture is a unique belief in Chinese society. The belief in gods and goddesses arises when people face unpredictable natural or man-made disasters, or events beyond their ability, and seek spiritual support in the hope that the gods will bless them and their families to be safe, secure, and even prosperous [5]. Wude Temple was founded in 1955 and has a history of more than 30 years. It has become a famous temple of wealth on both sides of the Taiwan Strait [6].

Because of the frequent transmission of cultural deeds, the temple has won the trust of the faithful. In less than half a century, more than 6000 branches have been established on both sides of the Taiwan Strait, making the belief in the god of wealth one of the most rapidly developing beliefs in Taiwan [7]. Since 2010, the current authorities have combined the concept of cultural and creative industries to transform the operation of the temple with an innovative commercial management model. Blue Ocean strategy, intelligent innovation, and online platforms are applied to adapt to the competitiveness. Facilities and activities such as robots, five-way gods of wealth cards, cafes, and the Triacademy attract more consumers [8]. Despite natural disasters in 2019–2020 and Taiwan’s overall economic downturn, the Lunar New Year Festival attracted more than 100,000 people [9]. Successive national holidays have brought in tens of thousands of people. On average, the temple attracts at least 4 million worshippers each year [10], indirectly creating more than a million business opportunities. This shows that Wude Temple has established itself in the hearts of the Taiwanese people and has become an indispensable part of their faith, bringing new opportunities for economic development to Beigang, which was originally an agricultural area.

Although religious beliefs and culture are specific to a region and have a unique appeal, they can attract people’s interest to experience or participate in them, leading to tourism or consumer behavior and indirectly generating the flow of people and capital. However, while tourism development is a major contributor to the economic development of villages, there are always oversights in management decisions and can have positive and negative impacts on the economy, society, and the environment [11,12,13], affecting local sustainable development.

Moreover, the impact of tourism development is not instantaneous but requires time to prove, and usually occurs after the end of tourism activities [12,14,15]. Especially with the development of Internet technology and software technology [16], coupled with the impact of the COVID-19 epidemic [17], people have begun to change their behaviors and choices in tourism activities. In order to understand the changes generated by the development, exploring from the perspective of the residents can provide insight into the real state of local changes [18,19,20,21], and exploring from the experience of tourists can understand the real effectiveness and shortcomings of tourism development promotion [21,22,23]. Religious and cultural promotion of tourism development can unite society, give people spiritual support, promote economic circulation, and improve the community environment, but it can also cause an increase in local social events, inflate consumer costs, leave behind waste, and cause air and environmental pollution. Therefore, in order to achieve sustainable village development, we must not capture the views of a single target group but must ensure that both residents and visitors have a basic understanding of environmental literacy and a consensus on sustainable development, in order to achieve the goal of promoting sustainable economic development in villages through religious culture. By exploring the development dilemma from both residents’ and visitors’ perspectives, not only can we obtain a more nuanced view of the problem [18,22,23], but we can also obtain a consensus between them to solve the di-lemma they face.

Furthermore, according to the literature in the National Digital Library of Theses and Dissertations in Taiwan, although the current research on religion, culture, and tourism are mostly qualitative in terms of investigating cultural characteristics and assets [18,19,20,21,22,23,24,25,26], followed by the awareness of religious activities [2,16,27], and cultural creativity and merchandise [28], the most quantitative research is on the impact of religious and cultural tourism [29,30]. However, there are only two studies on the Wude Temple of the God of Wealth in Beigang, Yunlin, and only qualitative studies on religious culture [5] and temple business model [8], and no other studies have been conducted to investigate the impact of religious culture development on local tourism development.

Therefore, the main purpose of this study was to understand the impact of religious and cultural tourism activities on the development of rural communities and the surrounding environment. From the perspective of environmental perceptions of people from different backgrounds, the study aimed to present the views on the impact of development on the current situation of local communities and the surrounding environment after the promotion of tourism activities with cultural resources in rural areas, to identify the shortcomings of development, and to propose suggestions for improvement towards the goal of common prosperity.

### 1.1. The Importance of Environmental Literacy to the Development of Cultural Tourism

Cultural tourism is the act of using cultural artifacts, historical relics, and ancestral cultural creations as resources to attract tourists to travel and spend money. Culture is an inseparable tourism asset for the tourism industry [31]. However, cultural tourism cannot be properly developed without a beautiful natural environment, convenient traffic planning and transportation, and sales services of related industries and commodities in the vicinity [32]. It is clear that the promotion of cultural tourism still requires the integration of local economic, social, and environmental resources, and joint planning and development in order to effectively promote cultural tourism.

However, tourism development cannot be achieved overnight as it requires public recognition and cooperation for effective planning and development [18,22,23]. There has been a long-standing positive and negative debate on tourism development [11,12,13], which has not yet been properly resolved. The main reason is that tourist travel or consumption behavior has an impact on local economic, social, and environmental conditions, while residents continue to change the existing economic, social, and environmental conditions in order to obtain rich rewards and improve their quality of life [23]. As the global environment becomes more and more degraded and the problems arising from tourist behavior become more and more serious, individuals and society recognize the interaction between their living environment and the surrounding natural environment and the need to focus on individual or collective solutions to present or future environmental problems [33] and, therefore, begin to advocate environmental education and to appreciate its value deeply.

The value of environmental education is to enhance people’s environmental awareness and sensitivity, knowledge of environmental concepts, environmental values and attitudes, environmental action skills, as well as environmental action experiences [34]. In addition to the goals of technological integration, proactive participation in the problem-solving, balanced world and local perspectives, sustainable development, and international cooperation [35], environmental literacy should be cultivated so that citizens have basic environmental values and can effectively judge the strengths and weaknesses of development and help improve the current situation to achieve sustainable development. This shows that although culture may be damaged by tourism development if people can improve their knowledge of environmental education, be sensitive to tourism development, and develop environmental literacy, they will be able to reduce the negative impacts of tourism development and achieve the goal of sustainable development.

### 1.2. Establishment of Environmental Literacy Helps People Develop Cultural Tourism

Tourism is a global industry and a major economic source, but with global climate change, the problem of carbon emission and waste pollution from tourism activities is becoming more and more serious, so governments have started to pay attention to this problem actively [36]. The best way to solve the problems caused by tourism development is to improve the direction of development decisions and raise the level of environmental literacy of the people [37] so that decision-makers and the public can move toward a sustainable attitude toward tourism development decisions. It can be seen that exploring the current state of tourism development with people’s current attitude toward environmental literacy is a good way to examine the effectiveness and shortcomings of sustainable development of tourism decisions.

Tourism is generally seen as an important means of promoting local economic development [10], increasing local employment opportunities, improving local infrastructure, tax revenues, foreign investment, etc., and thus attracting more industries to the area [38], which not only contributes to the local economy but is also very beneficial to the economic positioning of the area [39]. The economic impact is easier to measure, has a more robust methodology, and is more convenient and reliable in terms of the amount of data available, and the economy is also the core interest of tourism development policy [40], so the issue of economic impact has been emphasized earlier than social and environmental impact.

The economic impact can be examined in terms of the price of people, industrial construction, and village development [16], which can lead to entrepreneurship and employment opportunities, increased wage income, increased tourism construction, increased tourism industries, the integration of local specialty industries, increased leisure opportunities, integrity of public facility maintenance, tourism development feedback to the community, convenience of public transportation, increased local health standards, development protection policy settings, development of creative goods and increased expenditure costs, and increased land and housing prices [16,38,39,40,41]. Therefore, the researcher believes that the most accurate economic impact factors can be obtained by examining employment, wages, consumption, construction, industry, facilities, prices, incentives, health, cultural and creative activities, community feedback, and policy coordination.

The social impact is brought about by the intervention of tourism development, which can positively promote cultural and lifestyle communication, reduce population outflow, and maintain a more robust social structure, as well as contribute to the preservation of local culture due to the importance of tourism; a tendency for social relations to become increasingly indifferent and self-serving, and the negative effects are the change in the local social system, the possible deviation of individual behavior, the growing coldness and utilitarianism of social relations, and the local social conflicts due to racial discrimination [42]. These include improved material living conditions, diversification of occupational structures, decreasing trends of out-migration, narrowing of racial barriers, increased community openness, increased community conflicts, and seasonal unemployment generation and crime [43]. It will also influence the popularity of tourism, improve the quality of local tourism services and activities, increase leisure opportunities, encourage participation in community tourism affairs, provide sufficient local tourism indicators and options for recreational facilities, strengthen tourism development organizations, attract young people to return to their hometowns, preserve indigenous cultures, raise expenditure costs, increase land and housing prices, highlight local architectural features, make visitors feel friendly, interact well with residents, and increase cultural exchanges across the strait, and provide sufficient police and security personnel, and increase the willingness of people to revisit or purchase property in the area [20,21,44]. Therefore, the researcher believes that the most accurate social impact factors can be obtained by looking at tourism facilities, community building, living atmosphere, cultural security, and then exploring the aspects of popularity, service and activity quality, policy participation, tourism organization planning, cultural and architectural characteristics, security maintenance, community building, and public interaction.

There are two sources of environmental impacts, the first is the impact of the tourism activity itself, and the other is the impact of the facilities provided for the tourism activity [45]. The physical environment can be divided into the human-made environment and the natural environment, including soil erosion, vegetation destruction, and ecosystem changes [21,46]. The impact of the man-made environment includes traffic congestion, noise, and garbage caused by the increase of population, and the lack of space and environment resulting in the overload of physical facilities [16,20], and the impact of a large number of new era buildings forming an incongruous landscape with the existing facilities [43]. Therefore, researchers believe that the most accurate environmental impact factors can be obtained by looking at tourism and leisure facilities, natural ecosystems [21,22,23], public transportation, parking and open space, environmental quality of tourists, garbage, motor vehicle fumes, water, and air quality.

### 1.3. Analyze the Importance of the Relationship between Perceptions of Tourism Shocks and Willingness to Re-Tourism to Establish Rural Health and Environmental Sustainability

Tourism development can promote the local economy, enhance the living conditions and quality of life of local residents, and improve existing facilities and infrastructure to increase tourists’ willingness to visit and spend money there [23,38,40]. A good experience of the effectiveness of decision-making and development will help residents to actively cooperate and generate the will to continuously promote participation in tourism decision-making [47,48,49,50], and a good tourism consumption experience will also increase tourists’ willingness to participate in tourism [41,42,43,44,45,46,47,48,49,50,51,52,53]. The perceptions of decision effectiveness [54,55] and the current status of rural tourism development may also vary among different rights holders, genders, and ages [38,54,56,57].

Based on the above arguments, it is concluded that since villages can promote local economy through tourism development, improve community environment and facilities, enhance tourism conditions, and improve the quality of services and facilities, it will have a certain influence on the current situation of residents’ quality of life and tourists’ willingness to travel. Therefore, the researcher believes that there is a correlation between tourism impact perception and the desire to revisit or purchase a property.

## 2. Research Methods

### 2.1. Research Process and Framework

This study was designed to Beigang Wude Temple as a case and investigates the effect of Taiwan’s religious and cultural tourism for the development of rural tourism. Firstly, we collected relevant literature and conducted a questionnaire survey targeting local residents and tourists in Beigang from December 2020 to January 2021. A total of 800 questionnaires were distributed, and 600 valid questionnaires were retrieved, with a return rate of about 75%. The data were statistically analyzed using SPSS 22.0 software, and then descriptive analysis was conducted. Based on the analysis results, the field survey method was used to collect field information, and the interview method was used to collect the opinions of experts, seniors, and travelers, and the research paper was constructed by the sequence of summarization, organization, and analysis [20]. Finally, the multivariate verification analysis method was used to integrate the information of different research subjects, research theories, and methods, and to obtain accurate knowledge and meanings by comparing the research results from multiple perspectives and multiple data [20,58,59].

According to the above-mentioned literature [12,13,14,15,16,17,18,19,20,21,22,23,24,25,26,27,28,29,30,31,32,33,34,35,36,37,38,39,40,41,42,43,44,45,46,47,48,49,50,51,52,53,54,55,56,57,58], the study investigated the economic, social, and environmental-related aspects and issues from the perception perspectives of different backgrounds, and the specific research framework is shown in Figure 1.

According to the above framework, the research hypothesis is:

**Hypothesis** **1** **(H1).**
*The development of religious and cultural tourism has no significant impact on the current economic development of the village.*


**Hypothesis** **2** **(H2).**
*There is no significant impact of religious and cultural tourism development on the social development of villages.*


**Hypothesis** **3** **(H3).**
*There is no significant impact of religious and cultural tourism development on village environment development.*


**Hypothesis** **4** **(H4).**
*There was no significant correlation between the impact of economic development and the willingness to revisit.*


**Hypothesis** **5** **(H5).**
*There was no significant correlation between the impact of social development and the willingness to revisit.*


**Hypothesis** **6** **(H6).**
*There was no significant correlation between environmental development impacts and the desire to revisit.*


### 2.2. Research Tools

With reference to the literature on tourism impact [16,21,55,56,57], the economic, social, and environmental dimensions were categorized, and the subcomponents of the economic, social, and environmental dimensions were differentiated. The cognitive scale was designed using a five-point Likert scale, with a score of 5 for strongly agree, 4 for agree, 3 for generally agree, 2 for disagree, and 1 for strongly disagree, with the higher the score, the higher the cognitive level, and vice versa.

Reliability analysis can examine whether the measurement tool is reliable and stable. The α reliability coefficient method was used in this survey questionnaire, and SPSS 22.0 software was used to analyze the reliability of the questionnaire. Meanwhile, the coefficient value of Cronbach’s α is between 0 and 1, and the larger the α value, the better the correlation and the higher the reliability [60]. In general, an α value below 0.6 indicates that the internal consistency of the questionnaire is poor, an α value between 0.6 and 0.8 indicates that it is good, and if it is greater than 0.8, it indicates that the internal consistency of the questionnaire is very good [61]. The analysis showed that the Cronbach’s α coefficient was greater than 0.8 for economic, social, and environmental dimensions, so the reliability of the study questionnaire was higher for the economic dimension, as shown in Table 1.

### 2.3. Research and Analysis

The purpose of this study was to investigate the impact of religious culture on the development of village tourism in Beigang Wude Temple. Quantitative research can get the opinions of most people, but cannot get detailed questions [62,63]. Although qualitative research can only represent the suggestions of a small number of people, with the answers provided by representative people, deeper and subtle insights can be obtained [64]. Mixed research methods can make up for shortcomings [65]. In order to obtain the most factual suggestions for improvement, the study first asked five industry members, scholars, and citizens who are familiar with the local development process and have relevant professional backgrounds and created an outline of the interviews by referring to the issues on which at least three people had a consensus. Based on the results of the questionnaire analysis, we then applied focused interviews to seek the opinions of professionals, scholars, and citizens who are familiar with the local development process and have relevant professional backgrounds, to obtain more factual truths and construct the best recommendations as shown in Table 2.

Lastly, field surveys and interviews were conducted to collect actual information, and after summarizing, organizing, and analyzing all the data, a multivariate review was conducted.

The survey started in 2020, and the initial visitors were distributed all over the country. Due to limitations in manpower, material resources, and funding, field surveys were conducted first to observe the current status of village development and residents’ opinions. In addition, factors such as local farming, young people working outside the village, and the fact that the COVID-19 epidemic was not yet under control limited the initial collection of samples. Although the information was subsequently collected through a combination of online questionnaire platforms, the information collected by the researcher was flawed due to differences in respondents’ cooperation and proficiency in using 3C products. The limitations of the study will be presented in this paper, and we encourage subsequent researchers to correct them to improve on the study.

## 3. Results and Analysis

### 3.1. Background Analysis

The analysis revealed that there was not much difference in the status of the respondents (45.5% of residents and 54.5% of tourists), but most of them were women (40.9%), aged between 21 and 50 (74.2%), mainly residents of central (48.5%) and southern (42.4%) areas, and mostly used their own cars for transportation (89.4%). Most of the spending amount was less than 35.71 USD (75.8%), mainly for prayers, donations, joss paper, and incense (72.8%), as shown in Table 3.

### 3.2. Analysis of the Awareness of the Impact of Religious and Cultural Tourism on Village Development

Culture is an indispensable trace of human civilization, and faith is a source of inspiration for most people. Religious and cultural concepts of the immutability and equality of all beings have been the means of transmitting the correct social values and fostering environmental and cultural awareness in our country from ancient times to the present [57]. Therefore, based on the premise of environmental awareness, exploring people’s views on promoting cultural tourism and maintaining the overall environment of rural communities [34,66,67] can be a sound proposal for sustainable rural development. However, development has positive and negative impacts on the economic, social, and environmental levels [12,16], and different backgrounds may lead to different perspectives, and acquiring different perspectives is beneficial for obtaining the best suggestions for improvement [16,23,55,56,57].

The questionnaire was designed based on the literature, and a Likert scale was used, with 1 meaning strongly disagree and 5 meaning strongly agree. The basic statistical tests were used to explore people’s perceptions of the current status of village development, and then the t-test and ANOVA tests were used to explore the perception differences among different status, gender, and age, and then the interview information was compared and explored in a multivariate verification method [20,58].

#### 3.2.1. Analysis of the Awareness of the Impact of Religious and Cultural Tourism on the Village Economy

It was found that most people believed that the development of cultural tourism in Wude Temple has combined with local specialty industries (4.24), increased entrepreneurship and employment opportunities (3.95), and indirectly improved the standard of medical and health care (4.09). However, the effectiveness of the existing tourism development in giving back to the community (3.53) was not perceived, and the quality of public facilities (4.11) and public transportation (3.18) remained poor, which is not entirely consistent with the literature [14,21,53,54,55]. In addition, there was a significant difference in the perception of the current status of public facilities maintenance in communities with feedback from tourism development (*p* < 0.01), and residents felt worse about the effectiveness of public facilities maintenance; the older they were, the worse they felt, as shown in Table 4. Based on the above description, Hypothesis 1 was not confirmed.

Although environmental education has been included in the basic curriculum of Taiwan national education for many years, and students have been cultivating a sense of environmental conservation for many years, coupled with the Chinese culture’s promotion of the concept of benevolence and love, and the religion’s promotion of the awareness of equality of all beings, Taiwanese people are well aware of ecological and environmental conservation. However, the overall economic development of rural areas is insufficient, and all industries are still waiting to be developed. Although the people have the awareness of ecological and environmental conservation, they still hope to continue to develop rural cultural tourism activities by combining local religious customs (4.24), ecological environment, and agricultural products, so as to attract believers and tourists to visit the villages and create a large number of business opportunities, and to improve employment and entrepreneurship opportunities (3.95), medical and health care, as well as to improve the quality of life (4.09). However, since most of the existing public temple cultural institutions in Taiwan are private organizations with self-funded operations, and in order to avoid suspicion, officials seldom take the initiative to communicate with each other on development planning issues, and the degree of cooperation is low, as a result, the feedback received by villages is not effective (3.53), and the quality of public facilities (4.11) and public transportation remains poor (3.18). As a result, most people feel strongly about changes in the integration of special industries, entrepreneurship and employment opportunities, and the standard of medical and health care, but feel poorly about the effectiveness of the development of feedback villages, public transportation, and public facilities.

While the public has a wealth of environmental awareness and experience, policies need to be discussed, decisions need to be driven by human and resources, and results need to be proven over time so visitors who stay for a short period will not be able to accurately judge the difference between before and after changes. Moreover, as tourism development extends over time, the magnitude of change increases and only those who have lived here for a long time will be able to feel it deeply. Therefore, residents believe that the development of tourism does not give back to the community (residents < tourists; 3.77:4.39) and maintain public facilities (residents < tourists; 3.60:4.25), and the older they are, the worse they feel (20 under > 21–30 > 31–40 > over 51 > 41–50). Based on the above description, the analysis results obtained cannot be in line with the Institute of Research Hypothesis 1.

#### 3.2.2. Analysis of the Awareness of the Impact of Religious and Cultural Tourism on Village Society

It was found that most people thought that the development of cultural tourism in Wude Temple was helpful in enhancing the visibility of local tourism (4.47), friendly interaction between residents and tourists (3.91), and preservation of unique village humanistic architecture or landscape totems (4.17). However, the result is not entirely consistent with the literature [16,23,55,56,57], as police, firefighters, and security personnel (3.47), as well as tourism indicators (4.05), are not well planned, and architectural features (3.3) are gradually disappearing. The results are in line with the literature [16,55,56,57]. The issue of open space is significantly divided by gender (*p* < 0.01), and men believe that parking and open space facilities need to be improved, while people aged 31–40 are more sensitive to the issues of social participation and development of tourism organizations, as shown in Table 5. Based on the above description, Hypothesis 2 was not confirmed.

Most of the villages are remote areas where crowds do not easily gather. Promoting tourism development with religious, agricultural, and cultural specialties can effectively enhance local visibility (4.47). While residents expect to improve their quality of life and achieve long-term development, they do not want to lose their existing living habits (3.91), cultural cus-toms and features, and tourists do not want to lose their original village style and tourism features (4.17). However, due to the aging population, the outflow of young people (3.47), the small size of the village, and the limited space available for consumption (4.05), a large number of modern entertainment and consumption facilities have been built to meet the needs of tourists, forcing the demolition of existing buildings (3.3). Therefore, most people think that the visibility, preservation of unique village architecture or landscape totems, and interaction between residents and tourists are effective, while tourism indicators and police, firefighters, and security personnel are insufficient, and architectural features are gradually disappearing.

Since most people in Taiwan are highly educated and nurtured by sound professional knowledge and environmental teaching, they possess basic knowledge and have a high degree of environmental awareness and sensitivity. Most of the believers and cultural tourism tourists are family tourists, and there is little unused space in the rural areas. Excluding the living space of the existing residents, the shopping areas and stalls occupy the area, and the parking spaces for tourists are chaotic, resulting in tourists who are mainly male (father or elder brother) drivers often face the difficulty of finding a parking place (female > male; 3.56:3.97). Therefore, male citizens are more sensitive to the issue of parking and leisure facilities, while citizens aged 31–40 are more sensitive to the issue of social participation and the development of tourism organizations (31–40 > 20 under > over 51 > 21–30 > 41–50). Based on the above description, the analysis results obtained cannot be in line with the Institute of Research Hypothesis 2.

#### 3.2.3. Analysis of the Awareness of the Impact of Religious and Cultural Tourism on the Village Environment

It was found that most people thought that the cultural tourism development of Wude Temple helped preserve the historical scenery and relics (4.00) and that the temple authorities provided sufficient space for public toilets (4.00). However, the planning of transportation outside the temple (3.06) was inconvenient, and the public trash cans were not clearly set up and insufficient (3.39), which is not exactly the same as in the literature [16,23,55,56,57]. Although respondents of different status and gender had the same opinion, the older they were, the worse they felt about the planning of public toilets and the effectiveness of historical scenery and maintenance of monuments, as shown in Table 6. Based on the above description, Hypothesis 3 was not confirmed.

Although the ecological environment, history, customs, and ancient architecture are important tourism resources, the unique local religious culture is also a unique rural tour-ism feature. However, rural villages are scattered and have little space for tourism development, and the number is small and dense so maintaining resources (4.00) and providing a good public environment and sanitary space (4.00) is the key to improving the quality of tourism and services as well as sustainable development. However, because the main tourist activity space is far away from the road outside, the residential houses and the surrounding stores are crowded (3.06), the activity area is narrow and the available space for planning is limited (3.39). Therefore, most people think that the historical scenery and historical sites are well maintained, and the public toilets are well planned, but the public garbage cans are not well set up and insufficient in number, and the transportation is inconvenient.

Nevertheless, due to the limited building space available in the village temples, the lack of public space around them, the intermingling of residential and tourist areas, the aging population, the proliferation of elderly tourists, the outflow of young people, and the loss of labor force, it is impossible to provide adequate and complete public toilet facilities to satisfy the elderly worshippers (tourists). Moreover, the longer historical scenery and relics exist, the more precious they become, but the more easily they are damaged. Changes in scenery or resources can be experienced and felt by people who have lived there for a long time in a different time and context (20 under > 21–30 > 31–40 > 41–50 > over 51). Therefore, the older people are, the more deeply they feel that public restrooms are inadequate and that historical landscapes and monuments are not well maintained. Based on the above description, the analysis results obtained cannot be in line with the Institute of Research Hypothesis 3.

### 3.3. Correlation Analysis of Village Development Impact and Perception of Re-Tourism or Property Purchase Intention

Ultimately, tourism development aims to promote village development, improve existing facilities and infrastructure, meet the needs of tourists, and promote sustainable visitation and tourism consumption [23,38,40,55], which are sustainable development goals. Therefore, it is important to investigate the impact of village development on perceptions and willingness to revisit or purchase property to understand the key factors of people’s willingness to revisit or purchase properties. Therefore, the Pearson correlation analysis was used to examine the correlation between the impact on cognition and the willingness to revisit or purchase properties.

#### 3.3.1. Correlation between Economic Impact and Perception of Re-Tourism Willingness

It was found that there was a significant relationship between industrial development, private property prices, community development and the willingness to revisit or purchase properties (*p* < 0.001), and the effectiveness of industrial development (0.686), community development (0.618), and private property prices (0.588) influenced the willingness to recommend friends and relatives to travel and experience, and the results were not identical in the literature [23,38,40,55], as shown in Table 7. Based on the above description, Hypothesis 4 was no confirmed.

Although tourism development is currently advocated to be environmentally friendly and to move toward sustainable tourism development, for villages with religious culture and agricultural industry as development resources, having adequate industrial development, sound community development, and stable prices for people’s livelihoods are still the main keys to attracting people. Based on the above description, the analysis results obtained cannot be in line with the Institute of Research Hypothesis 4.

#### 3.3.2. Correlation between Society Impact and Perception of Re-Tourism Willingness

It was found that there was a significant correlation (*p* < 0.001) between community building, the culture of life, cultural security and the willingness to revisit or purchase property, and the effectiveness of community building (0.686), cultural security (0.618), and culture of life (0.588) affected the willingness to recommend friends and relatives to travel and experience, and the results were not identical in the literature [19,34,36,51], as shown in Table 8. Based on the above description, Hypothesis 5 was no confirmed.

Since villages have limited space for tourism development, organizing volunteers or bringing in manpower to maintain culture and law and order, deeply cultivating local human customs and cultural characteristics, using existing space for community building, as well as proper planning of tourism highlights, will be the basis for people to recommend their friends and relatives to visit the villages. Based on the above description, resulting in the analysis results obtained cannot be in line with the Institute of Research Hypothesis 5.

#### 3.3.3. Correlation between Environment Impact and Perception of Re-Tourism Willingness

A significant correlation (*p* < 0.001) was found between village environment, public health, and the willingness to revisit or purchase properties, and the effectiveness of public health (0.752) and village environment (0.317) influenced people’s attractiveness and willingness to re-engage in local activities and the results were not identical in the literature [23,38,40,55], as shown in Table 9. Based on the above description, Hypothesis 6 was no confirmed.

The environment and sanitary conditions of tourism are the main factors for people to consider in their travel activities, especially in the current poor travel environment surrounded by viruses, a safe and sanitary travel environment is a key consideration. Therefore, maintaining a clean and safe public sanitary space in the village environment is a key factor to attract tourists to visit again and recommend their friends and relatives to visit with them. Based on the above description, the analysis results obtained cannot be in line with the Institute of Research Hypothesis 6.

## 4. Conclusions

Survey results show that although cultural tourism helps villages to improve their reputation, preserve historical sites, increase the integration of special industries, promote interaction among people, increase entrepreneurship and employment opportunities, and improve the standard of public toilets and medical sanitation, problems such as the lack of tourism feedback, inadequate village development, low number of public garbage cans, unclear settings, inconvenient transportation, insufficient public facilities, tourism indicators, and police and fire safety personnel, and the disappearance of local architecture have yet to be solved. 

It was concluded that creating parking spaces, providing a comfortable resting place for tourists, creating an open exchange of ideas, and raising public awareness and consciousness of the environment would increase the importance of public issues such as village visibility, citizen interaction, ancient architecture, culture and totems, public health and transportation, and entrepreneurial development in the village, as well as address the concerns of local residents and some men and people over the age of 31–40. It will also improve community building and security, enrich cultural resources, provide adequate industrial infrastructure and development, stabilize prices, and achieve a safe and sanitary public environment, thus increasing the desire of people to revisit and making the village a recommended destination for family travel, and achieving the goal of sustainable development of rural environment and health.

Based on the above results, the following suggestions are made:

### 4.1. Local Government

Development does not only depend on local rural characteristics and tourism resources but also requires administrative and financial support from government agencies in order to have proper development space and community planning.

If the local government can entrust experts and scholars to conduct field Tacha, reforming surrounding tourist moving lines, link temples and cultural organizations, to collect the views of residents, aid to promote the depth of cultural tourism.

### 4.2. Local Authorities

Development does not rely solely on government resources and enthusiastic public input, but also on talents with professional knowledge and skills, in order to continuously explore local characteristics, create uninterrupted tourism appeal, and achieve sustainable development goals.

If local governments or tourism development organizations can refer to the suggestions of local cultural organizations and professionals, conserve local cultural assets, and use resources to develop tourism activities or products, they can create tourism highlights.

### 4.3. Policy Makers

Visionary leaders are critical but gathering more information and recruiting more expertise can lead to innovative, sound, and trend-aligned decisions.

If government agencies can invite existing villagers, organize community volunteer organizations, set up entrepreneurial technology courses, encourage residents to participate, and combine local high school courses and manpower, it can solve the problem of insufficient development manpower.

### 4.4. Suggestions for Future Research

Since the study mainly takes Beigang Wude Temple as a case study, it explores the influence of Taiwan’s religious and cultural tourism on the development of rural tourism, and the different cultural customs and characteristics of different regions may also cause different village development impacts. 

Therefore, the researchers believe that it is recommended that future researchers continue to explore religious and cultural tourism or related issues based on differences in different countries, regions, different rights, and ages, and understand the understanding and differences of peers on this topic in order to improve related research flaws.

## Figures and Tables

**Figure 1 ijerph-18-02731-f001:**
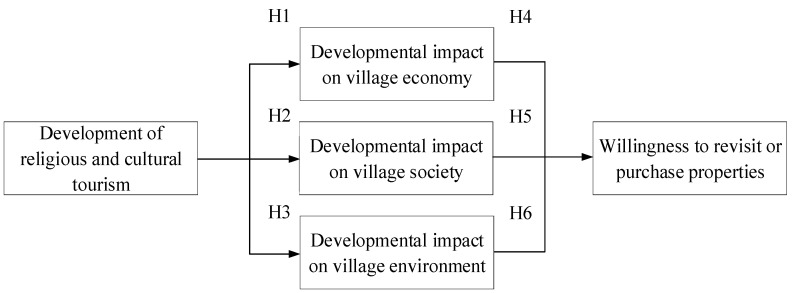
Study framework.

**Table 1 ijerph-18-02731-t001:** Classification table of tourism impact dimensions.

Construct		Dimension	
Basic Variables		gender (male/ female), identity (residents: tourists) age (20 down: 21–30: 31–40: 41–50: 51 up) Place of residence (North, Central, South, East, Outlying Islands) Mode of coming to worship (motorcycle, car, bus, cab or rental, bicycle, walking) Spending amount (within 1000, 1001–5000, above 5001) Main consumption content (prayers, incense burning, cultural and creative items, snacks, parking fees, tea, accommodation, or other)	
construct		dimension	cronbach’s α
Economy	Industry infrastructure (0.903)	Increasing tourism infrastructure Increasing the tourism industry Integration of local specialty industries Maintenance of public facilities	0.900–0.903
	Cost of living (0.918)	Increasing entrepreneurship and employment opportunities Increasing salary income Rising rent/land and housing prices	0.899–0.918
	Village development (0.908)	Tourism development giving back to the community Improving public transportation Improving medical and health standards Development of protection policies Development of creative products	0.900–0.908
Society	Community building (0.905)	Enhancing tourism awareness Improving service quality Improving the quality of tourism activities Increasing leisure opportunities Promoting community participation Adequate tourism indicators	0.900–0.905
	Living atmosphere (0.900)	Development of tourism organizations Youth development in their hometowns Preservation of unique village architecture or landscape totems Loss of architectural features	0.886–0.900
	Cultural security (0.910)	Friendly interaction between tourists and residents Adequate police, firefighters, and security personnel Willingness to revisit or purchase properties in the area Sufficient parking and resting facilities	0.899–0.910
Environment	Village environment (0.941)	Environmental quality affected by tourists Convenient transportation Oil and smoke pollution from motorcycles Noise pollution Excessive tourist waste	0.905–0.941
	Public sanitation (0.970)	Clearly set up and sufficient number of public garbage cans Adequate maintenance measures for historical scenery and monuments Adequate number of public toilets	0.898–0.914
Willingness to revisit		Happy to participate in any activities again Willing to recommend friends and family to visit Be willing to suggest improvements to the place Willing to share travel experience	0.932–0.970

**Table 2 ijerph-18-02731-t002:** Background information of the interviewees and outline of the interview.

Identity	Gender	Residence Time/Years of Work Experience	Identity	Gender	Residence Time/Years of Work Experience
Tourist guide	Male	40	senior citizens	Male	30
Tourist guide	Female	25	senior citizens	Female	45
professor	Male	15			
construct	issues				
impact of tourism development	On the premise of sustainable development of the environment, please answer the following questions based on your environmental literacy and current perception, based on research and surveys: 1. What problems may exist between the promotion of religious and cultural tourism activities and the current economic, social, and environmental development of the village? How to solve it? 2. What causes the economic, social, and environmental problems? How to improve? 3. What causes the economic, social, and environmental development to affect tourists’ willingness to travel or purchase?				

**Table 3 ijerph-18-02731-t003:** Background disguised analysis table.

Gender	%		Place of Residence	%
male	40.9%		north	9.1%
female	59.1%		central	48.5%
age	%		south	42.4%
20 under	18.2%		eastern/islands	0%
21–30	24.2%		Contents of consumption	%
31–40	24.2%		Praying and donation	36.4%
41–50	25.8%		Joss paper, and incense	36.4%
Over 51	7.6%		Creative cultural souvenirs of Wude Temple	4.5%
travel mode	%		Tea and drinks	1.5%
Bicycle	7.6%		Meals and snacks	13.6%
Walking	1.5%		Accommodation or other expenses	4.5%
Personal Motorcycle	43.9%		Parking fee	3.0%
Private car	45.5%		Spending amount	%
Bus or tour vehicle	1.5%		35.71 under	75.8%
identity	%		Over 35.75	24.2%
resident	45.5%			
tourist		54.5%		

**Table 4 ijerph-18-02731-t004:** Analysis of the awareness of the impact of religious and cultural tourism on the village economy.

Construct	Dimension	μ	Identity			Gender			Age						
			Residents	Tourists	*p*	Male	Female	*p*	20 down	21–30	31–40	41–50	51 up	*p*	Post Hoc
Industry infrastructure (4.17)	Increasing tourism infrastructure	4.17	4.17	4.17	0.56	4.15	3.18	0.57	4.17	4.44	4.38	3.82	3.8	0.51	-
	Increasing the tourism industry	4.17	4.2	4.14	0.69	4	4.28	0.18	4.17	4.56	4.38	3.71	3.8	0.6	-
	Integration of local specialty industries	4.24	4.23	4.25	0.65	4.26	4.23	0.06	4.33	4.44	4.5	3.76	4.2	0.41	-
	Maintenance of public facilities	4.11	3.77	4.39	0.01 *	4.15	4.08	0.58	4.5	4.38	4.38	3.41	3.8	0.00 *	20 under > 21–30 > 31–40 > over 51 > 41–50
Cost of living (3.63)	Increasing entrepreneurship and employment opportunities	3.95	3.83	4.06	0.32	4.07	3.87	0.5	4.25	4.06	4.31	3.35	3.8	0.23	-
	Increasing salary income	3.85	3.37	4	0.27	4.15	3.64	0.84	3.92	4.13	4.19	3.29	3.6	0.48	-
	Rising rent/land and housing prices	3.78	3.37	3.67	0.21	3.41	3.62	0.86	3.17	3.69	3.81	3.29	3.8	0.87	-
	Tourism development giving back to the community	3.53	3.2	3.17	0.13	3.15	3.21	0.96	2.75	3.13	3.19	3.59	3	0.8	-
Village development (3.96)	Improving public transportation	3.18	3.83	4.31	0.08	4.11	4.08	0.97	4.33	4.25	4.31	3.53	4.2	0.04	-
	Improving medical and health standards	4.09	3.47	4.14	0.02	4	3.72	0.95	4.17	4	4.06	3.24	3.8	0.03	-
	Development of protection policies	3.83	3.6	4.25	0.01 *	4.04	3.9	0.74	4.33	4	3.81	3.06	4.2	0.02	-
	Development of creative products	3.79	4.13	4.14	0.02	4.15	4.13	0.23	4.33	4.31	4.06	3.18	4.2	0.32	-

* *p* < 0.01.

**Table 5 ijerph-18-02731-t005:** Analysis of the awareness of the impact of religious and cultural tourism on village society.

Construct	Dimension	μ	Identity			Gender			Age						
			Residents	Tourists	*p*	Male	Female	*p*	20 down	21–30	31–40	41–50	51 up	*p*	Post Hoc
Community building (4.2)	Enhancing tourism awareness	4.47	4.37	4.56	0.22	4.44	4.49	0.21	4.25	4.5	4.75	4.35	4.4	0.12	-
	Improving service quality	4.20	3.97	4.39	0.02	4.19	4.21	0.12	4.33	4.19	4.5	3.82	4.2	0.08	-
	Improving the quality of tourism activities	4.17	3.9	4.39	0.13	4.15	4.18	0.23	4.33	4.13	4.5	3.76	4.2	0.15	-
	Increasing leisure opportunities	4.23	4.1	4.33	0.36	4.22	4.23	0.79	4.25	4.19	4.63	3.88	4.2	0.02	-
	Promoting community participation	4.11	3.87	4.31	0.17	4.19	4.5	0.77	4.25	4.06	4.5	3.76	3.8	0.00 *	31–40 > 20 under >over 51 > 21–30 > 41–50
	Adequate tourism indicators	4.05	3.97	4.11	0.25	4	4.08	0.31	4.08	4.13	4.44	3.76	3.4	0.02	
Living atmosphere (3.84)	Development of tourism organizations	4.08	3.93	4.19	0.21	4.01	4.1	0.08	4.25	4.06	4.38	3.71	4	0.01 *	31–40 > 20 under > 21–30 > over 51 > 41–50
	Youth development in their hometowns	3.77	3.43	4.06	0.04	3.74	3.79	0.29	4.17	3.94	4.31	2.88	3.6	0.41	-
	Preservation of unique village architecture or landscape totems	4.15	3.93	4.33	0.09	4.11	4.18	0.35	4.25	4.31	4.38	3.65	4.4	0.22	-
	Loss of architectural features	3.30	3.3	3.31	0.93	3.22	3.36	0.04	2.92	3.69	3	3.53	3.2	0.97	-
Cultural security (3.71)	Friendly interaction between tourists and residents	3.91	3.8	4	0.21	3.85	3.98	0.96	4.33	3.94	4.13	3.35	4	0.05	-
	Adequate police, firefighters and security personnel	3.47	3.43	3.5	0.77	3.33	3.56	0.3	3.92	3.75	3.88	2.65	3	0.02	-
	Willingness to revisit or purchase properties in the area	3.85	3.93	3.78	0.39	3.7	3.95	0.13	4.25	3.88	4.06	3.41	3.6	0.14	-
	Sufficient parking and resting facilities	3.80	4.2	3.47	0.12	3.56	3.97	0.01 *	4.08	4.06	4	3.41	3	0.09	-

* *p* < 0.01.

**Table 6 ijerph-18-02731-t006:** Analysis of the awareness of the impact of religious and cultural tourism on the village environment.

Construct	Dimension	μ	Identity			Gender			Age							
			Residents	Tourists	*p*	Male	Female	*p*	20 down	21–30		31–40	41–50	51 up	*p*	Post Hoc
Village environment (3.62)	Environmental quality affected by tourists	3.67	3.93	3.44	0.57	3.56	3.74	0.23	3.33	3.81	3.63		3.76	3.8	0.37	-
	Convenient transportation	3.06	2.87	3.22	0.56	2.74	3.28	0.19	3.92	3.19	3.06		2.47	2.6	0.55	-
	Oil and smoke pollution from motorcycles	3.73	3.73	3.72	0.62	3.33	4	0.5	3.67	3.75	3.94		3.53	3.8	0.55	-
	Noise pollution	3.88	3.93	3.83	0.94	3.78	3.95	0.32	3.83	3.88	3.94		3.88	3.8	0.13	-
	Excessive tourist waste	3.77	3.83	3.72	0.77	3.63	3.87	0.17	3.58	3.63	3.81		3.94	4	0.29	-
Public sanitation (3.79)	Clearly set up and sufficient number of public garbage cans	3.39	3.63	3.19	0.85	3.19	3.54	0.53	4	3.69	3.56		2.59	3.2	0.09	-
	Adequate maintenance measures for historical scenery and monuments	4	3.9	4.03	0.02	3.93	4	0.24	4.42	4.13	4.25		3.41	3.4	0.00 *	20 under > 21–30 > 31–40 > 41–50 > over 51
	Adequate number of public toilets	4	4.43	3.64	0.22	3.74	4.18	0.09	4	4.19	4.63		3.47	3.2	0.00 *	20 under > 21–30 > 31–40 > 41–50 > over 51

* *p* < 0.01.

**Table 7 ijerph-18-02731-t007:** Correlation analysis of the economic impact and re-tourism intention.

Issue	Industry Infrastructure	Cost of Living	Village Development
Happy to participate in any activities again	0.672 **	0.547 **	0.610 **
Willing to recommend friends and family to visit	0.686 **	0.588 **	0.618 **
Be willing to suggest improvements to the place	0.684 **	0.541 **	0.574 **
Willing to share travel experience	0.633 **	0.506 **	0.621 **

** *p* < 0.001.

**Table 8 ijerph-18-02731-t008:** Correlation analysis of society impact and re-tourism intention.

Issue	Community Building	Living Atmosphere	Cultural Security
Happy to participate in any activities again	0.748 **	0.700 **	0.764 **
Willing to recommend friends and family to visit	0.777 **	0.714 **	0.767 **
Be willing to suggest improvements to the place	0.724 **	0.671 **	0.736 **
Willing to share travel experience	0.767 **	0.672 **	0.711 **

** *p* < 0.001.

**Table 9 ijerph-18-02731-t009:** Correlation analysis of the environmental impact and re-tourism intention.

Issue	Village Environment	Public Sanitation
Happy to participate in any activities again	0.317 **	0.752 **
Willing to recommend friends and family to visit	0.320 **	0.744 **
Be willing to suggest improvements to the place	0.305 *	0.724 **
Willing to share travel experience	0.303 *	0.706 **

* *p* < 0.01. ** *p* < 0.001.

## Data Availability

No data support.

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
