# Peer review of "Research on the Development of Religious Tourism and the Sustainable Development of Rural Environment and Health"

_ijerph, 2021, doi:10.3390/ijerph18052731_

Round 1

Reviewer 1 Report

Thank you for the opportunity to read the paper “Uncover the illusion of tourism development and seek a sustainable consensus to promote the development of religious tourism and the overall rural environment and health”.

Please see my comments below.

  • Line 21 – the authors stated that “….to statistically validate 600 samples”. Shouldn’t “samples” be replaced with subjects/ participants/ respondents?
  • I suggest the authors to present in Introduction the main objectives of the paper, and eventually the conceptual framework  (dimensions and items) and the relationships among constructions.
  • 174-179 – several aspects are repetead (“the negative impact is the change of the local social system, which may lead to deviations in personal behavior, (....), and The negative effects are the change in the local social system, the possible deviation of individual behavior, ....”) – please correct.
  • Table 1 is not very clear. Does that classification indicate the constructions/ dimensions which were obtained through a Principal Component Analysis? Or how did the authors grouped the issues? Do the issues mean variables/ items that were included in the questionnaire? Please clarify this table and the concepts included.
  • Lines 281-282 – the authors indicate that “A questionnaire survey was used to analyze 385 samples from residents and tourists”. The authors mentioned earlier that the sample included 600 subjects. Why did they use the term “samples” instead of subjects/ participants/ respondents?
  • Table 2: there is a mistake (“cender” instead of gender) – please proofread; what do the authors mean by “the elderly” in Table 2?
  • What is the relation between the qualitative research and the quantitative research? Is the quantitative research related to the results of the qualitative research or vice versa? Also, I suggest the authors to separate these two researches into different sections in order to better emphasize the role/ contribution/ results of each research.  
  • Line 289 – there is a little mistake at the year 2020 – please proofread.
  • Table 4 is not clear at all. Is this table a contingency table? Do the authors make a comparison between male and female/ residents and tourists? If yes, how were these groups compared?
  • Lines 343-371; 388-410; 423-443: is this discussion related to the results of the analysis? If yes, please provide some results/ figures that indicate/ demonstrate these results.
  • In the Literature section, the authors presented a theoretical framework and some connections among constructions. Are these relationships significant and positive/ negative? Are the hypotheses confirmed/ rejected? What are the results that support/ reject these hypotheses?
  • Section 5 should be improved. Please indicate what results of the analysis support the suggestions from this section. Who will benefit from these results? How could these results be used by the decision factors?

Good luck!

Author Response

Reviewer 1

  • Line 21 – the authors stated that “….to statistically validate 600 samples”. Shouldn’t “samples” be replaced with subjects/ participants/ respondents?
  • Dear reviewers: Thank you for your suggestion, we have made corrections. Such as line 21.
  • I suggest the authors to present in Introduction the main objectives of the paper, and eventually the conceptual framework  (dimensions and items) and the relationships among constructions.
  • Dear reviewers: Thank you for your suggestion, and we will add it. Such as lines 101-103.
  • 174-179 – several aspects are repetead (“the negative impact is the change of the local social system, which may lead to deviations in personal behavior, (....), and The negative effects are the change in the local social system, the possible deviation of individual behavior, ....”) – please correct.
  • Dear reviewers: Thank you for your suggestion, we have made changes. Such as lines 173-176.
  • Table 1 is not very clear. Does that classification indicate the constructions/ dimensions which were obtained through a Principal Component Analysis? Or how did the authors grouped the issues? Do the issues mean variables/ items that were included in the questionnaire? Please clarify this table and the concepts included.
  • Dear reviewers: Thank you for your suggestion, we have made changes. As shown in Table 1.
  • Lines 281-282 – the authors indicate that “A questionnaire survey was used to analyze 385 samples from residents and tourists”. The authors mentioned earlier that the sample included 600 subjects. Why did they use the term “samples” instead of subjects/ participants/ respondents?
  • Dear reviewers: Thank you for your suggestion, we have made changes. Such as lines 281-282.
  • Table 2: there is a mistake (“cender” instead of gender) – please proofread; what do the authors mean by “the elderly” in Table 2?
  • Dear reviewers: Thank you for your suggestion, we have made changes ("cender" has changed to "gender", "the elderly" has changed to "senior citizens"). Such as Table 2.
  • What is the relation between the qualitative research and the quantitative research? Is the quantitative research related to the results of the qualitative research or vice versa? Also, I suggest the authors to separate these two researches into different sections in order to better emphasize the role/ contribution/ results of each research.  
  • Dear reviewers: Quantitative research can get the opinions of most people, but can't get detailed questions (PA Ochieng ,2009; Berry, N., Lobban, F. & Bucci, S., 2019). Although qualitative research can only represent the suggestions of a small number of people, with the answers provided by representative people, deeper and subtle insights can be obtained (B Cypress, 2018). Mixed research methods can make up for shortcomings (N Walliman, 2017). Therefore, the researchers believe that the use of two methods can find a more appropriate solution for the research.
  • Line 289 – there is a little mistake at the year 2020 – please proofread.
  • Dear reviewers: Thank you for your suggestion, we have made changes. Such as line 289.
  • Table 4 is not clear at all. Is this table a contingency table? Do the authors make a comparison between male and female/ residents and tourists? If yes, how were these groups compared?
  • Dear reviewers: Thank you for your suggestions. The presentation of Form 4 has been adjusted, and the content analysis has been explained in lines 331-337.
  • Lines 343-371; 388-410; 423-443: is this discussion related to the results of the analysis? If yes, please provide some results/ figures that indicate/ demonstrate these results.
  • Dear reviewers: Thank you for your suggestions. The text of Lines 343-371; 388-410; 423-443 has been supplemented with corresponding data.
  • In the Literature section, the authors presented a theoretical framework and some connections among constructions. Are these relationships significant and positive/ negative? Are the hypotheses confirmed/ rejected? What are the results that support/ reject these hypotheses?
  • Dear reviewers: Thank you for your suggestions. The suggestions have been explained in the manuscript and the text description has been completed.
  • Section 5 should be improved. Please indicate what results of the analysis support the suggestions from this section. Who will benefit from these results? How could these results be used by the decision factors?
  • Dear reviewers: Thank you for your suggestion. We will revise the content of Section 5 and have more accurately stated the research insights.

Finally, once again sincerely thank you for your suggestions and make the manuscript more perfect.

Reviewer 2 Report

Thank you for your research study. I have read your work and have several comments. While I am not impressed about quality of the work (particularly, its theoretical contribution and practical contribution), I think the paper can be improved significantly.  

First, the title is too long. Please mention a brief title that sufficiently represents your research.

When I read the abstract I felt the authors have done research on a lot of things and everything. Please clarify the goals of the study. Also, the abstract is very vague. What samples? Suggestions about what? What do you mean by improving police and fire security personnel (do you mean improving their service delivery)?

The introduction is weekly written. The goals of the research are unclear. The authors need to be more upfront about the goals.

In the introduction section, there is a lot of information about the case study, which can be moved to the methodology section. Also, there is no reference or support for some of the claims. For example, see the sentence below. Why would readers trust a claim if it is a personal statement?

“Instead, because of the uniqueness of local religious beliefs and culture, they are recognized by the public and attract more believers to worship them, which in turn adds to the mystery of local culture and makes local religious beliefs and culture more valued and preserved by the public.”

Also, not sure what you mention “Despite this year's natural disaster…” in the introduction section. What if the reader reads your paper next year?

Not sure why you chose reviewing theses and dissertations in the National Digital Library of Theses and Dissertations to explain gap in research. Why not reviewing academic journal and conference publications that are relevant to this research topic as well?

I skipped reading your literature review because the sub-sections in the literature review sound separate. What is or are the theme(s) in your literature review? Why those sub-headings in the literature review section? the literature review section doesn’t sound coherent. Please write them in a coherent way. You may also want to add a text at the start of the literature review section that connects those sub-headings together and to the overall goal of the research.

Under research methods you say “The purpose of this paper is to investigate the impact of Beigang Wude Temple religious culture on rural tourism development.” If this is your research goal, it is too specific and narrow. But if the temple is your case for your case study, it’s fine. Please revise the goal.

The authors mention that they conducted survey and interviews and suddenly mention hypotheses. This is a very week hypothesis development and explanation. Currently, it feels like each hypothesis has happened magically. It reads like: “I know what I say. Just trust me”. Please provide all the details of your data collection (e.g. participant demographics of survey and interviews) in an upfront way and also clearly mention why you conducted a survey and then interviews. How the survey items were developed. These need to be mentioned in an upfront and clear way.

Also, it is important to provide more details about your interview questions. What were the questions and how the questions were developed?

It is also very important to clarify/mention the percentage or rate of agreement among coders of qualitative data? Did at least two of the authors agree on the codes/themes? If yes, what is the percentage of agreement (e.g. is it 100%, 80%, or…?)? Use this reference (which is the most relevant and recent reference) in your paper to answer the question:

Nili, A., Tate, M., Barros, A., & Johnstone, D. (2020). An approach for selecting and using a method of inter-coder reliability in information management research. International Journal of Information Management54, 102154.

I am not a survey expert, so I do not comment on the analysis part.

I have never heard “Conclusion and Suggestion” section. Do you mean Conclusion and Discussion? In the discussion part, you can mention theoretical contributions and provide recommendations for practice (explain practical contribution). The suggestions (recommendations) need more details. Also, why suggestion for future research has been treated the same way as suggestions for practice? You need to explain suggestions for future research as a different paragraph which is standalone.  

I hope the authors use the feedback to improve their work.

Author Response

Reviewer 2

First, the title is too long. Please mention a brief title that sufficiently represents your research.

 Dear reviewers: Thank you for your suggestion. We have adjusted the content of the topic. Such as line 1-2

When I read the abstract I felt the authors have done research on a lot of things and everything. Please clarify the goals of the study. Also, the abstract is very vague. What samples? Suggestions about what? What do you mean by improving police and fire security personnel (do you mean improving their service delivery)?

Dear reviewers: Thank you for your suggestion. We have adjusted the abstract content. Such as lines 22-28.

 The introduction is weekly written. The goals of the research are unclear. The authors need to be more upfront about the goals.

 Dear reviewers: Thank you for your suggestion, we have added it. Such as line 96-98

In the introduction section, there is a lot of information about the case study, which can be moved to the methodology section. Also, there is no reference or support for some of the claims. For example, see the sentence below. Why would readers trust a claim if it is a personal statement?

“Instead, because of the uniqueness of local religious beliefs and culture, they are recognized by the public and attract more believers to worship them, which in turn adds to the mystery of local culture and makes local religious beliefs and culture more valued and preserved by the public.”

Dear reviewers: Thank you for your suggestion. We have added the literature to support the research theory. Such as References 1-2.

 Also, not sure what you mention “Despite this year's natural disaster…” in the introduction section. What if the reader reads your paper next year?

 Dear reviewers: Thank you for your suggestion. We have revised the manuscript description. Such as lines 55-56.

Not sure why you chose reviewing theses and dissertations in the National Digital Library of Theses and Dissertations to explain gap in research. Why not reviewing academic journal and conference publications that are relevant to this research topic as well?

 Dear reviewers: Thank you for your suggestion. In addition to the oriental religious and cultural documents related to the manuscript, we have added many international journals to increase the credibility of the research arguments.

I skipped reading your literature review because the sub-sections in the literature review sound separate. What is or are the theme(s) in your literature review? Why those sub-headings in the literature review section? the literature review section doesn’t sound coherent. Please write them in a coherent way. You may also want to add a text at the start of the literature review section that connects those sub-headings together and to the overall goal of the research.

 Dear reviewers: Thank you for your suggestion. After discussion, we revised the sequential numbering of the titles in the manuscript and adjusted the title naming of the documents, hoping to improve the visualization of the manuscript.

Under research methods you say “The purpose of this paper is to investigate the impact of Beigang Wude Temple religious culture on rural tourism development.” If this is your research goal, it is too specific and narrow. But if the temple is your case for your case study, it’s fine. Please revise the goal.

 Dear reviewers: Thank you for your suggestion. After discussion, we have adjusted the narrative method of the research case object. Such as lines 221-223.

The authors mention that they conducted survey and interviews and suddenly mention hypotheses. This is a very week hypothesis development and explanation. Currently, it feels like each hypothesis has happened magically. It reads like: “I know what I say. Just trust me”. Please provide all the details of your data collection (e.g. participant demographics of survey and interviews) in an upfront way and also clearly mention why you conducted a survey and then interviews. How the survey items were developed. These need to be mentioned in an upfront and clear way.

 Dear reviewers: Thank you for your suggestion. We have added the theoretical basis for selecting interview subjects and research methods. Such as lines 275-284.

Also, it is important to provide more details about your interview questions. What were the questions and how the questions were developed?

 Dear reviewers: Thank you for your suggestions. The formation of issues and the interview process have been explained in lines 275-284.

It is also very important to clarify/mention the percentage or rate of agreement among coders of qualitative data? Did at least two of the authors agree on the codes/themes? If yes, what is the percentage of agreement (e.g. is it 100%, 80%, or…?)? Use this reference (which is the most relevant and recent reference) in your paper to answer the question:

 Dear reviewers: Thank you for your suggestion to supplement the explanation. On lines 279-285.

Nili, A., Tate, M., Barros, A., & Johnstone, D. (2020). An approach for selecting and using a method of inter-coder reliability in information management research. International Journal of Information Management54, 102154.

 Dear reviewers: Thank you for your suggestions. We have referred to the literature and adjusted the presentation of the data in the manuscript.

I am not a survey expert, so I do not comment on the analysis part.

 I have never heard “Conclusion and Suggestion” section. Do you mean Conclusion and Discussion? In the discussion part, you can mention theoretical contributions and provide recommendations for practice (explain practical contribution). The suggestions (recommendations) need more details. Also, why suggestion for future research has been treated the same way as suggestions for practice? You need to explain suggestions for future research as a different paragraph which is standalone.  

 Dear reviewers: Thank you for your suggestions. Section 5 presents a summary of the main findings of the research. The discussion and practical action suggestions required for the analysis results have been explained in lines 346-378, 394-418, 432-455, 476-482, 492-498, 508-514, 546-549 etc.

Finally, once again sincerely thank you for your suggestions and make the manuscript more perfect.

Reviewer 3 Report

How to make tourism sustainable has been increasingly important. This paper also contributes to the development of tourism policy, however, in addition to the concrete example, other existing practices at other religious sites could be also involved.

The literature review is of high quality, authors are able to have multiple approach to the development of tourism and its relationship to local development. The methodology is properly used. The conclusions are based on the resulst, however, I would suggest to include the approach of the local government, local authorities, policy makers, because it is not enough if there is feedback from visitors, but such issues must be also discussed with the local decision-makers as well.

There are some minor suggestions regarding the text:

Line 14: „…Managemen” a „t” has to be added to the end of the word

Line 289:”…202010…” needs to be formulated as commonly used 2020 October – as it is assumed

Line 304: „…1,000 NT…” has to be converted into e.g. USD to be able to understand by most of the readers

line 343: „…of our national education…” – „our” needs to be changed to the country, plural must be avoided

Author Response

Reviewer 3

The literature review is of high quality, authors are able to have multiple approach to the development of tourism and its relationship to local development. The methodology is properly used. The conclusions are based on the resulst, however, I would suggest to include the approach of the local government, local authorities, policy makers, because it is not enough if there is feedback from visitors, but such issues must be also discussed with the local decision-makers as well.

There are some minor suggestions regarding the text:

Line 14: „…Managemen” a „t” has to be added to the end of the word

Dear reviewers: Thank you for your suggestion, it has been revised. As in line 14.

Line 289:”…202010…” needs to be formulated as commonly used 2020 October – as it is assumed

Dear reviewers: Thank you for your suggestion, it has been revised. As in line 289(291).

Line 304: „…1,000 NT…” has to be converted into e.g. USD to be able to understand by most of the readers

Dear reviewers: Thank you for your suggestion, the calculation has been revised to US dollars. Such as line 307.

line 343: „…of our national education…” – „our” needs to be changed to the country, plural must be avoided

Dear reviewers: Thank you for your suggestion, it has been modified into a narrative style. Such as lines 346-347.

Finally, once again sincerely thank you for your suggestions and make the manuscript more perfect.

Round 2

Reviewer 1 Report

Dear Authors,

The paper was much improved. However, I have still some methodological concerns, which will be further detailed:

  • Table 1: It’s still unclear why do you use terms as Classification or Issue, which are not usually used in the statistical literature? What do you mean by Classification? Is it construct/ dimension? If yes, did you use a Principal Component Analysis to classify the variables? Please clarify. What do you mean by issue? Is it variable/ item?
  • What do you mean by “Above findings and hypothesis X is not exactly the same”? I think the wording is not correct from a statistical point of view.
  • Tables 4, 5, 6: why wouldn’t you split the identity column in two columns (including the two groups) in order to be more understandable? The same is for gender. In addition, the Age column is unclear. Therefore, I suggest the authors to modify/ rearrange this column in order to be easier to understand by readers.  Again, why wouldn’t you replace the word issue with variable/ item in order to be statistically correct?
  • What do you mean by “Based on the above description, resulting in the analysis results obtained can not be in line with the Institute of Research Hypothesis X”? Why wouldn’t you better say “Based on the above description, Hypothesis X was no confirmed/ was rejected”?
  • Section 4: why wouldn’t you compile the subsections 4.1, 4.2 etc. in only one section?

Good luck!

Author Response

Table 1: It’s still unclear why do you use terms as Classification or Issue, which are not usually used in the statistical literature? What do you mean by Classification? Is it construct/ dimension? If yes, did you use a Principal Component Analysis to classify the variables? Please clarify. What do you mean by issue? Is it variable/ item?
..
Dear reviewer
Thanks for your suggestion, we have revised the wording in the form. 
..
    What do you mean by “Above findings and hypothesis X is not exactly the same”? I think the wording is not correct from a statistical point of view.
    Tables 4, 5, 6: why wouldn’t you split the identity column in two columns (including the two groups) in order to be more understandable? The same is for gender. In addition, the Age column is unclear. Therefore, I suggest the authors to modify/ rearrange this column in order to be easier to understand by readers.  Again, why wouldn’t you replace the word issue with variable/ item in order to be statistically correct?
    What do you mean by “Based on the above description, resulting in the analysis results obtained can not be in line with the Institute of Research Hypothesis X”? Why wouldn’t you better say “Based on the above description, Hypothesis X was no confirmed/ was rejected”?
...
Dear reviewer
Thank you for your suggestion, we will adjust the presentation of the table and modify the presentation in the text. 
...
    Section 4: why wouldn’t you compile the subsections 4.1, 4.2 etc. in only one section?
Dear reviewer
Thanks for your suggestion, Chapter 4 has been adjusted. 
